# Correlation between the Positive Effect of Vitamin D Supplementation and Physical Performance in Young Male Soccer Players

**DOI:** 10.3390/ijerph19095138

**Published:** 2022-04-23

**Authors:** Michał Brzeziański, Monika Migdalska-Sęk, Aleksandra Czechowska, Łukasz Radzimiński, Zbigniew Jastrzębski, Ewa Brzeziańska-Lasota, Ewa Sewerynek

**Affiliations:** 1Department of Endocrine Disorders and Bone Metabolism, Medical University of Lodz, 90-752 Lodz, Poland; michal.brzezianski@umed.lodz.pl (M.B.); ewa.sewerynek@umed.lodz.pl (E.S.); 2Academic Laboratory of Three-Dimensional Anthropometry, Medical University of Lodz, 92-213 Lodz, Poland; 3Department of Biomedicine and Genetics, Medical University of Lodz, 92-213 Lodz, Poland; ewa.brzezianska@umed.lodz.pl; 4Academic Laboratory of Movement and Human Physical Performance, Medical University of Lodz, 92-213 Lodz, Poland; 5Department of Physiology and Biochemistry, Gdansk University of Physical Education and Sport, 80-336 Gdansk, Poland; lukaszradziminski@wp.pl (Ł.R.); zb.jastrzebski@op.pl (Z.J.)

**Keywords:** vitamin D supplementation, 25(OH)D, skeletal muscle power, VO_2_max, football, soccer players

## Abstract

The aim of this study was to determine whether supplementation with vitamin D during eight weeks of high-intensity training influences muscle power and aerobic performance in young soccer players. A total of 25 athletes were divided into two groups: the supplemented group (GS; *n* = 12; vitamin D 20,000 IU, twice a week) and the non-supplemented group (GN; *n* = 13). A set of measurements, including sprint tests, explosive power test, maximal oxygen uptake (VO2max), and serum 25(OH)D concentration, were obtained before (T1) and after (T2) the intervention. A significant group x time interaction was found in the 25(OH)D serum levels (*p* = 0.002; ES = 0.36, large). A significant improvement in VO2max was found in the TG (*p* = 0.0004) and the GS (*p* = 0.031). Moreover, a positive correlation between 25(OH)D and VO2max (R = 0.4192, *p* = 0.0024) was calculated. The explosive power tests revealed insignificant time interactions in the average 10-jump height and average 10-jump power (*p* = 0.07, ES = 0.13; *p* = 0.10, ES = 0.11, respectively). A statistically insignificant trend was observed only in the group-by-time interaction for the sprint of 10 m (*p* = 0.05; ES = 0.15, large). The present study provides evidence that vitamin D supplementation has a positive but trivial impact on the explosive power and locomotor skills of young soccer players, but could significantly affect their aerobic performance.

## 1. Introduction

It has been assessed that 80–100% of cholecalciferol (vitamin D) in humans should be produced in the skin, in the keratinocytes from 7-dehydrocholecalciferol (7-DHC), after exposure to sunlight radiation (280–315 mm UVB). Under the influence of the absorbed energy, 7-DHC undergoes transformation to previtamin D3, and in the next step via thermoconversion, to vitamin D3. During biochemical processes, vitamin D is converted into 25-hydroxyvitamin D (25(OH)D) in the liver and then to 1,25-dihydroxyvitamin D (1,25(OH)2D) in the kidneys [1]; 25(OH)D and 1,25(OH)2D circulate in the blood mostly bound to vitamin D-binding protein (DBP). After being released from DBP to the tissues, 1,25(OH)2D triggers, through the intracellular vitamin D receptor (VDR), numerous metabolic actions in the tissues and organs [1,2]. The pleiotropic effect of vitamin D is related to the fact that vitamin D receptors (VDRs) have been found in almost all human cells, including those in bones and muscles [3]. It has been recognized that 1,25(OH)2D acts through both genomic and non-genomic actions to regulate the expression of vitamin D-responsive genes (over 900 gene variants) [4,5].

So far, many scientific studies have been published worldwide that demonstrate a variety of health benefits from vitamin D, including its wide spectrum of skeletal and extra-skeletal activities [6,7,8]. In myotubes or osteoblasts, vitamin D can act, dependent on or independent of its binding to the VDR [9,10]. Moreover, in vitro and in vivo studies performed on animals showed that vitamin D modulates muscle cell proliferation and differentiation [9,11]. Furthermore, interactions of vitamin D with cells in vivo in adult rats are associated with the promotion of cell proliferation, differentiation, and the regenerative process in skeletal muscle [12]. Moreover, it was documented on animal models that vitamin D affected the size and number of fast-twitch muscle fibers [13,14]. Vitamin D also regulates muscle contractile function [15]. It is suggested that the status of the bioactive form of vitamin D and the expression of VDR in muscle cells seem to be key factors involved in calcium-binding efficiency for muscle fiber twitch [16]. The mechanism by which vitamin D may improve aerobic performance remains unclear. It is possible that the CYP (cytochrome P450) enzymes involved in the activation of vitamin D to the biologically active 1,25(OH)2D contain heme proteins that may affect the oxygen binding to hemoglobin. Through this mechanism, the vitamin D, via its conversion, may improve oxygen transport and aerobic performance [17]. Vitamin D may also play a role in the iron regulation of the body, improving hemoglobin production [18]. Vitamin D also may improve the function of mitochondria, influencing cellular respiration, oxygen capacity, and finally, aerobic performance [19].

It is commonly known that vitamin D deficiency in the global population in Europe is ubiquitous [20]. Hence, the problem of vitamin D deficiency and its adequate supplementation represent an important issue in public health and clinical practice. It may be associated with its well-known regulation of calcium, phosphate absorption, and pleiotropic actions in organs and tissues in the body. Cholecalciferol is the most common preparation used as supplementation and treatment of vitamin D deficiency in Poland and Europe, unlike in the USA, where ergocalciferol (D2) is largely used [21,22].

A high prevalence of vitamin D deficiency has been recorded in various groups of professional athletes during regular high-intensity training [16,23,24,25,26,27]. This almost pandemic deficiency is believed to be related to intensive physical activity, higher metabolic rate, frequent indoor training, and lack of sufficient exposure to sunlight [28,29]. Published studies confirmed that the risk of vitamin D deficiency in athletes increases at higher latitudes, in northern countries, and during winter [28,30,31]. However, the observed effect of vitamin D supplementation varied, depending on the age, sex, and basal status of vitamin D in studied athletes. Moreover, supplementation methods modalities, including the dose or scheme of vitamin D administration, have had inconclusive results [23,24,32,33,34,35,36]. 

Studies focusing on the impact of vitamin D supplementation on global muscle strength/power [33,34], physical performance [35,36], both indicators [24,32], and muscle injury prevention [35] in different groups of athletes are ambiguous. Stockton et al. found a high effectiveness of vitamin D supplementation on muscle strength [37]. Close et al. noted improvements in vertical jump height [23], similar to Wyon et al., which in addition, found an improvement in the isometric strength of classical ballet dancers [36].

On the other hand, some authors have found no evidence of any significant impact of vitamin D on the various indicators of physical performance [38,39,40]. Nieman et al. showed that a large dose of ergocalciferol (vitamin D2) supplementation in athletes during a 6-week period had no effect on muscle function [41]. In the meta-analysis conducted by Zhang et al., vitamin D supplementation positively affected lower limb muscle strength, but had no impact on overall muscle strength outcomes in athletes [34].

In many studies on athletes, a high prevalence of vitamin D insufficiency/deficiency was reported [28,30,42]. This situation is associated with many years of extensive physical indoor training along with a lack of ultraviolet exposure from sunlight. Previous results revealed that oral supplementation with vitamin D improves muscle strength/power [34] and muscular performance [23], and protects against muscle damage [33,36]. The conducted study clearly confirmed that variability in the vitamin D supplementation regimens used in young athletes can change the vitamin D status from deficient or insufficient to adequate. However, the optimal intake of 25(OH)D levels that ensure effectiveness for athletes has not yet been fully established [24,31,43]. Recently, it has also been noticed that, when the vitamin D status in serum is adequate, vitamin D supplementation does not cause additional benefits to muscle strength and power [44].

Moreover, the association of vitamin D and aerobic capacity has been previously studied, but the conclusions have been ambiguous. Athletes’ aerobic capacity (cardiorespiratory fitness) is determined by the efficiency of oxygen consumption, which depends on many factors, such as blood oxygen capacity, oxygen supply to tissues, cardiac output, lung diffusion capacity, and others. When all the oxygen supply functions reach high efficiency, the ability to perform exercises with a high load will increase, and the athlete will have higher physical efficiency [45]. The gold standard of measurement of aerobic performance is VO2max, which is the maximal volume of oxygen utilized by the muscles during exhaustive exercise per minute [46]. 

It has been well-documented that cardiorespiratory fitness reflects the ability of the body to work continuously for an extended period of time. Some data show a positive correlation between basic vitamin D serum level and aerobic performance [47,48] or between vitamin D serum concentration after vitamin D supplementation and aerobic performance during a training cycle [49]. On the other hand, a rather trivial effect of vitamin D supplementation on aerobic performance in athletes has also been observed [33,50,51,52]. Other authors claimed that the best way to recover physical performance is by plain and accurate training methods that improve the technical and motor skills of players [50,53,54].

However, due to the common 25(OH)D deficiency in athletes, the vitamin D supplementation is generally recommended [3,55,56]. Therefore, we assumed the hypothesis that vitamin D supplementation can be an important factor supporting the training process and advancing sports effectiveness. In light of the controversial evidence, the aim of our study was to determine whether supplementation with vitamin D (20 000 IU of cholecalciferol twice a week for 8 weeks) influences muscle strength/power and aerobic capacity in young soccer players.

## 2. Materials and Methods

### 2.1. Experimental Approach

The study was approved by the Medical University of Lodz Ethics Committee (RNN/283/18/KE, 18 September 2018), and the investigation was carried out following the rules of the Declaration of Helsinki of 1975. All players and their parents or legal guardians were provided with detailed information about the study procedures and gave their written consent. The training plan for the athletes was developed at the “Arka” Gdynia football club in Gdynia, Poland. Laboratory tests were carried out at the Department of Endocrine Disorders and Bone Metabolism and at the Department of Biomedicine and Genetics of the Medical University of Lodz.

### 2.2. Characteristics of the Study Group

Forty young soccer players initially took part in the research. However, due to injuries (*n* = 4), illnesses (*n* = 7), changes of club (*n* = 2), and other reasons (*n* = 2), not all of them met the inclusion criteria. Finally, twenty-five soccer players from an under-19 team participated in the study. All athletes had a valid athlete’s health card. Selected biological features of athletes and selected training characteristics are shown in Table 1.

The soccer players enrolled in the study were in a period of the training cycle that, during the experiment, included 7 units of training (Monday–Friday) and friendly games (Saturday). Each workout began with a warm-up. Beforehand, the athletes implemented the standard soccer training plan developed by the team’s training staff. Training loads carried out at the first stage of preparations (4 weeks) included low- and medium-intensity exercises (with a predominance) of aerobic and mixed (aerobic–anaerobic) metabolism. At the second stage of preparations (4 weeks), the training loads were aerobic–anaerobic and anaerobic. At this stage, high-intensity exercises based on the interval method (e.g., small games) and repetition (e.g., shaping the motion speed) were dominant. The training cycle was monitored based on training diaries, and training plans arranged by trainers. Only players who participated in at least 85% of training sessions and matches were considered in the final analysis.

The study inclusion criteria were: male gender and possession of an up-to-date athlete’s health card, i.e., they were free of any pathology or clinical conditions. Participants were also not taking medications two weeks before the data collection, and vitamin supplementation for one month before the study. Additionally, athletes had to follow a sports diet suggested by a club nutritionist and demonstrate 80% attendance during the experiment. 

The criteria for exclusion from the experiment were female gender, use of stimulants (e.g., caffeine) or drugs, smoking, current use of medications, history of chronic diseases, sports injuries during the experiment, low physical capacity resulting from training absenteeism or an injury one month before the experiment, and lack of athlete’s or their parents’ or legal guardians’ written consent. The competitors were informed about the purpose of the research project.

### 2.3. Vitamin D supplementation and Analysis

The examined athletes (total group; TG) were randomly divided into two groups: with vitamin D supplementation (GS) and without vitamin D supplementation (GN).
Group I (*n* = 12)—subjected to training and supplemented with vitamin D (cholecalciferol in a dose of 20,000 IU (Decristol, Sun-Farm Sp. z o.o., Lomianki, Poland), which was taken twice a week for 8 weeks). Supplementation was carried out under the supervision of a sports dietitian during the preparation for the annual training cycle (from January to March).Group II (*n* = 13)—subjected to training only, without vitamin D supplementation.

The dispensation procedure was double-blind. Players from the GS received pills with vitamin D content, while the GN players received placebo pills (pills with identical color and shape). To ensure double blindness, only the club coordinator was aware of the group assignment. Coaching staff and medical staff delivered the pills prepared by the coordinator to the players. Such an approach guaranteed that neither the players nor the coaches and physiotherapists had knowledge of which players were assigned to the GS and the GN. Furthermore, before statistical analysis, the groups were coded (GS—GR1; GN—GR2) to ensure that the researcher who calculated the data was not aware of group affiliation. Supplementation and testing were carried out in late winter when vitamin D synthesis was likely to be minimal. During the study, the subjects were asked several times whether they had taken the vitamins as instructed. All the subjects reported that they had followed the instructions. This was confirmed by the measurements of 25(OH)D before and 8 weeks after the supplementation. During the experiment, 70% of the players fed themselves in the school boarding house and 30% at home. The diet of the players was low in vitamin D. The estimated consumption during the experiment was about 150 UI/day for competitors eating at home and about 200 UI/day for competitors eating in the boarding school. This value was estimated on the basis of previously applied methods [57,58,59]. A calculator of vitamin D content in food products (Dieta 6.0 Software, Warsaw, Poland, 2018) obtained measurements seven days into the experiment. These calculations were based on the list of consumed products reported by the athletes.

### 2.4. Serum Collection and Evaluation of Serum 25-Hydroxyvitamin D Concentration

The collection of biological material was conducted by nurses according to reliable protocols of collection, transport, and storage of biological material. Five milliliters (5 mL) of whole arm vein blood was collected from each athlete in both groups (GN and GS) at baseline (time point T1) and at the end of the study, i.e., after 8 weeks of the training cycle and/or cholecalciferol supplementation (time point T2). All participants in the study had blood drawn at the same time and had the same interval after the last training session, i.e., the athletes did not perform any high-intensity training in the 70–72 h prior to the blood sample collection. For example, the players played a match (sparring), i.e., very high-intensity training, on Saturday at 12:00. The next day, Sunday, was completely free from training. On Monday, the players then participated in low-intensity recovery training from 1:00 p.m. to 2:00 p.m.; this was practically an “active rest” because the training included only low-intensity jogging and stretching exercises, i.e., low-intensity aerobic exercise with little effect on muscle fatigue. Finally, the blood sample was taken on Tuesday morning. The above blood sampling scheme applied to both T1 and T2 points; however, the first set of samples (T1) were taken in January and the second (T2) in March after 8 weeks of training and/or vitamin D supplementation.

The blood was collected in sterile tubes without anticoagulant. All tubes were marked with identification numbers. The blood samples were left to form a clot at room temperature (approximately 30–45 min). After centrifugation at 2400× *g* rpm for 10 min at room temperature, the serum (supernatant) was carefully separated from the clot into Eppendorf tubes. The obtained biological material was frozen at −20°C.

Following this, the 25(OH)D concentration was evaluated using a commercial kit “Elecsys Vitamin D Total II” (Roche Diagnostics GmbH, Mannheim, Germany). The level was read by ECLIA automated electrochemiluminescence on a COBAS e411 system. Reference values for 25(OH)D were: 30–70 ng/mL normal value; 20–30 ng/mL slight deficiency; 10–20 ng/mL moderate deficiency; <10 ng/mL severe deficiency [21,60]. The assay sensitivity was 4.01 ng/mL. The inter-assay coefficients of variation (CV) ranged from 2.20% to 10.70% across a working range of 3.00–70.00 ng/mL for four serum standards in 84 samples.

### 2.5. Sprint Tests

Sprint tests were performed indoors, i.e., in a sports hall on a running track surface. To avoid potential fatigue, the testing took place 72 h after the match. Before the test, all the players performed a 20-min warm-up involving dynamic exercise and accelerations. Each participant performed two runs at maximal intensity over distances of 5, 10, and 30 m. The best (shortest) times achieved at each distance were used in the final analysis. The run was timed using photocells (Smartspeed Timing Gates System Fusion Sport, Cooper Plains, Brisbane, Australia). The players started in a standing position, with their front leg on a line 0.6 m before the first photocell.

### 2.6. Jump Tests

The explosive power test was performed in the indoor hall using a tensometric mat (Smart Jump Mat 120 × 120 cm^2^—Fusion Sport, Cooper Plains, Brisbane, Australia). Before the test, the athletes performed a 20-min warm-up involving five vertical jumps. The test comprised two maximal vertical jumps without swinging arms (Squat Jump, SJ) and two with swinging arms (Counter Movement Jump, CMJ). The resting period between jumps was two minutes. Only the best, i.e., the highest, jump was used in the subsequent analysis. Then, after a 5-min break, the athletes made 10 jumps as high as possible, one after another without a rest break. The mean value of the 10 jumps was included in the performance analysis.

### 2.7. Multistage Shuttle Run Test

The multistage shuttle run test (MST) was performed on the indoor running track according to Leger and Lambert [61]. Due to acyclic specificity of movement, this test is commonly used to assess aerobic fitness in soccer players. The pace of the run was set by beep signals provided by a sound system. Participants ran between two lines spaced 20 m apart without any time for recovery. The time between the signals was shortened with each level. The initial running velocity was set at 8 km·h^−1^ and increased by 0.5 km·h^−1^ until exhaustion. The test was terminated when the player was not able to follow the signal for two consecutive shuttles. The total distance covered by the participants (number of shuttles × 20 m) was considered the result of the test and used in the statistical analysis [62]. Moreover, the maximal oxygen uptake (VO2max) was estimated using the formula proposed by Ramsbottom et al. [63].

### 2.8. Statistical Analysis

The data are presented as means ± standard deviations (SD). All statistical analyses were conducted using Statistica 13.0 (Tulsa, OK, USA, 2016). The data sets were analyzed using the Shapiro–Wilk test for normal distributions. A two-way mixed ANOVA was applied to analyze the between-group (GN vs. GS) and within-group (T1–T2) effects. If the interaction effects were found to be significant, the post hoc Tukey’s HSD test was performed. The partial eta squared (ηp2) was calculated to determine the effect size (ES), which was then classified as small (≥0.01), medium (≥0.06), and large (≥0.14) [64]. Additionally, the T1–T2 changes for the TG were calculated with the t-test for dependent variables. 

Furthermore, the potential relationships between 25(OH)D concentration and physical test results for the TG were tested using Spearman’s correlation coefficient. The level of correlation was fixed in the following categories: very strong (R ≥ 0.80), moderate (R = 0.60–0.79), fair (R = 0.30–0.59), and poor (R ≤ 0.29) [65]. The significance level was set at *p* < 0.05.

## 3. Results

The mean ± SD of 25(OH)D concentrations and physical fitness parameters (sprint 5 m, 10 m, 30 m, MST distance, VO2max, 10 jumps, SJ, and CMJ) at two time points (T1, T2) in each group of athletes (TG, GN, and GS) are presented in Table 2. 

All study groups demonstrated similar 25(OH)D concentrations in T1 (before the experiment): no significant difference was found between the supplemented and non-supplemented groups. Statistically significant differences in 25(OH)D level were observed between T1 and T2 in each group (GN and GS) (*p* = 0.002; ES = 0.36, large). In addition, 25(OH)D demonstrated a significant effect with regard to group (*p* = 0.006; ES = 0.29, large) and time (*p* = 0.0003; ES = 0.44, large). The concentration of 25(OH)D increased in all groups from T1 to T2: by 38% in the TG, by 6% in the GN, and by 70% in the GS. Statistically significant differences between T1 and T2 were observed in the TG (*p* = 0.0027) and the GS (*p* = 0.0002). The results of the post hoc analysis in the GS and the GN for 25(OH)D are given in Figure 1. 

The main effect of time was significant for such variables of physical fitness as MST distance and VO2max (*p* = 0.0001, ES = 0.42, large and *p* = 0.001, ES = 0.41, large, respectively). At T2, both MST distance and VO2max increased in all groups compared to T1; however, statistically significant differences between T1 and T2 were only found in the TG (*p* = 0.0004) and the GS (*p* = 0.031). The results of the post hoc analysis for VO2max in the GS and the GN are given in Figure 1. 

The analysis of the explosive power test, i.e., the 10 jumps test, revealed an insignificant increase in mean height and power. In addition, an insignificant, medium time interaction was found for the mean height of 10 jumps and an identical group interaction for the mean power of 10 jumps (*p* = 0.07; ES = 0.13, medium for both). Similarly, the SJ results increased from T1 to T2 in all groups; however, the difference was not significant. No significant group, time, or group × time interactions were found in the CMJ group. 

Regarding the running speed tests, no significant improvement was found between T1 and T2 with regard to the time for the 5 m, 10 m, or 30 m runs. However, an insignificant trend in the group-by-time interaction was observed for the 10 m sprint (*p* = 0.05; ES = 0.15, large).

The correlations between 25(OH)D concentration and physical fitness parameters (sprint 5 m, 10 m, 30 m, MST distance, VO2max, 10 jumps, SJ, and CMJ) are given in Table 3. A positive correlation was found between 25(OH)D and MST distance and 25(OH)D and VO2max (R = 0.4192, *p* = 0.0024; for both variables) (Figure 2). No significant correlations were found between 25(OH)D concentration and the remaining physical fitness parameters.

## 4. Discussion

It has been suggested that athletes who present serum vitamin D deficiency should receive supplementation in order to improve their performance [27]. Indeed, athletes display better performance in summer when sun exposure is more available, and the effects of naturally synthesized vitamin D (even with deficiencies) combined with supplementation have a noticeable effect on improving muscle function in the sprint test [66]. However, it is still unclear whether vitamin D supplementation can compensate for deficiency in all athletes, regardless of ethnic differences and individual biological characteristics. It is also unclear whether vitamin D supplementation truly improves muscle function and aerobic capacity, and the dose of vitamin D that should be recommended for athletes [3,34,38,51]. Indeed, despite the importance of this topic to the athlete population, very few studies have been published. Therefore, the present study examines the influence on supplementation in muscle explosive power (strength) and aerobic capacity in young soccer players. The explosive power as a mechanical power and muscle strength are considered to be highly important parameters during training, as they ensure effective performance [24]. Moreover, soccer players typically train with high-power exercises during the competition period, which usually lasts from September to the end of December, depending on the competition schedule. Many studies have documented a high frequency of vitamin D deficiency/insufficiency exactly during this period [23,24,26]. 

Hence, the first important observation of our work is that elite young athletes included in the study, despite the use of a diet designed for young athletes living in school dormitories, showed insufficient serum vitamin D level at baseline (mean 26.7 ng/mL ± 10.01) during the winter training cycle. Our observations are consistent with previous findings, which also confirmed vitamin D deficiency/insufficiency in athletes living in Poland [31,55,67,68,69]. Therefore, the recommendations regarding the necessity of vitamin D supplementation in the winter, and immediately afterwards, in Central European countries [31,55,67] seem correct, although a clear dose of vitamin D supplementation for the Polish sports population has not been established. Of course, the supplementation dose is dependent on many factors, such as diet, latitude of the place of residence, and training conditions. 

When selecting the dosage of vitamin D, we relied on the guidelines of Polish experts regarding the compensation of the deficiency or suboptimal concentration of said vitamin [70]. Our main goal was to obtain fast vitamin D saturation (optimal concentration of 25(OH)D in the highest percentage of the studied subpopulation and in a short period of time (8 weeks)). Compared to our study, the weekly administration of 30,000 IU vitamin D for 12 weeks was safe and effective for the normalization of 25(OH)D levels to the desirable level of >30 ng/mL in deficient patients [71]. In patients over 70 years old, annual doses as high as 500,000 IU for 3–5 yrs increased the risk of falls and fractures [72]. However, such adverse effects have not been observed in younger, healthy populations. In our study, the population consisted of adults and almost adults (age 17.5 +/− 0.7 years) for whom the recommended dosage of vitamin D is 4000–10,000 IU/24 h or 30,000–60,000 IU once a week [70]. Our dosage was equivalent to a dose of 40,000 IU/week for only 8 weeks. It has been examined that monthly high doses of 100,000 IU vitamin D supplementation taken over a median of 3.3 yrs did not increase kidney stone risk or serum calcium [73]. In addition, the recommendations for vitamin D deficiency management for Eastern and Central Europe that have recently been published suggest that higher doses of vitamin D (e.g., 6000 IU per day) may be used for the first 4 to 12 weeks of treatment, if a rapid correction of vitamin D deficiency is clinically indicated, before continuing with a maintenance dose of 800 to 2000 IU per day [74].

It should be noted that most previous studies on the use of vitamin D supplementation found it to have a positive effect on improving serum level [23,32,52,69]. Only a few studies have indicated that a large-dose vitamin D2 supplementation in athletes had no effect on total 25(OH)D [41]. In the present study, athletes used cholecalciferol supplementation at a dose of 20,000 IU twice a week for 8 weeks; this was sufficient to obtain a statistically significant increase in serum 25(OH)D levels by the end of the project. Other authors focusing on Polish soccer players observed similar results, despite using a different supplementation regimen [33,52,69], such as 2000 UI/day for 3 weeks [33] or 5000 UI/day for 8 weeks [40,63]. It has also been proposed that the daily dosage should be 5000 UI for athletes in the winter season [23]. Finally, meta-analysis findings suggest that 4–12 weeks of supplementation with vitamin D with a daily dosage over 2857 IU in winter can alleviate vitamin deficiency and increase serum 25(OH)D concentrations from insufficient to sufficient [32]. In the present study, 20,000 IU twice per week for 8 weeks also alleviated basal serum vitamin D deficiency in the tested soccer players. The biological half-life and the pharmacology of vitamin D indicate that it is suitable to take vitamin D for weekly or even monthly dosing [33,34,35,36]. It has been shown in some studies that daily 1000 IU, weekly 7000 IU, and monthly administrations of 30,000 IU of vitamin D3 provide equal efficacy and safety profiles [75]. Due to the long whole-body half-life of vitamin D, [76,77], daily dosing is generally considered unnecessary; therefore, monthly or weekly dosing seems ideal [75]. Even higher doses of vitamin D3 supplementation, i.e., 50,000 IU twice weekly for 5 weeks, was safe among patients with low 25(OH)D levels [78]. 

It has been documented that doses higher than 2000–3000 UI/day are sufficient to protect against muscle damage during intense workouts and to increase skeletal muscle function [3,22], whereas those lower than 1000 IU/day (a dose of 600–800 IU/day) may not be sufficient to compensate for intense physical training [79]. Interestingly, von Hurst PR et al. emphasize that interventions with vitamin D have a positive effect on muscle function only in participants with insufficient 25(OH)D levels, i.e., <50 nmol/L [49]. Thus, the optimal vitamin D supplementation scheme, including its dose and serum level, especially in the context of improving sports performance and muscle regeneration, is still debatable [80]. Such studies on athletes lack standardized measurement techniques, and it has been proposed that different muscle groups may respond differently to vitamin D supplementation [34,41]. 

The specific nature of soccer requires numerous acyclic efforts, such as sprints, turns, accelerations, decelerations, changes of direction, jumps, etc. Such activities depend on correct neuro-motor function, which can be improved by vitamin D supplementation by facilitating maximal power and speed [81]. The present study focuses on an evaluation of explosive muscle power (explosive strength) and power-related sprint performance in young athletes without and with vitamin D supplementation. Explosive power (strength) refers to the ability of a muscle to exert maximal force in a short period of time (e.g., jumping), which is commonly measured by vertical jump performance. The present study evaluated explosive power using the SJ (squat jump), CMJ (counter movement jump), and MST dist. (multistage shuttle run test). These tests play an important role in soccer training due to the high frequency of jumping during training and match play [82]. In the explosive power test results, an insignificant, medium time interaction was found for the mean height of 10 jumps and an identical group interaction for the mean power of 10 jumps. The SJ and CMJ results also increased but not on the level of significance during the analyzed period in all groups. Our results are similar to Jastrzębska et al., where it was demonstrated that supplementation with vitamin D of the group of soccer players have a positive but minor effect on explosive power/strength [69]. In this study, no significant difference in performance improvement was observed in the supplemented athletes and control group, despite a 119.2% increase in serum vitamin D level [69]. In turn, vitamin D supplementation was not found to influence the standing long jump test results [39]. It should be noted that it is difficult to compare the results of these studies as they use different tests on muscle function (viz. 1-RM bench press and leg press vs. vertical jump height). Moreover, in other studies, there is the possibility of not taking into account errors of measurement and temporal instability [82].

No significant improvement of power-related sprint results was observed in the present study. Vitamin D supplementation was only associated with the 10 m sprint running test result, and the change was not significant. This is consistent with previous findings by Close et al. [43]. Similarly, vitamin D supplementation was found to have a weak positive effect on power-related characteristics, such as running speed, despite a significant increase in serum vitamin D levels in the treatment group [69]. In addition, vitamin D level was not demonstrated to have any significant correlation with jump or sprint performance in a group of young female football players from Sweden [40]. In contrast, Bezuglov et al. observed in their study that supplementation was associated with a statistically significant increase in the results of the 5, 15, and 30 m sprint tests [39], but not of the standing long jump test. However, serum levels of vitamin D were directly associated with the performance in jump tests and in both the 10 and 20 m sprint tests in professional Greek soccer players [81].

In our study, it is important to note that vitamin D supplementation appeared to have a positive effect on the MST distance. The supplemented group covered a significantly longer distance after the training cycle, which was not observed in the non-supplemented group. These results are in line with those of a previous study conducted by Skalska et al.; however, the effect size in run distance (sprint) was not statistically significant [52]. Our results, as in other independent studies, indicate that vitamin D supplementation significantly improved serum levels, which may be related (not always significant) to the improvement in muscle power and speed.

A very important role in estimating physical performance is played by maximum oxygen uptake VO2max. This quantity is defined as the maximum oxygen consumption absorbed by the body in one minute during maximum exercise [83]. Aerobic capacity plays an important role in soccer and has a major influence on technical performance. Therefore, our present study evaluates the influence of vitamin D supplementation on VO2max. The findings indicate a significant increase in VO2max, as determined by MST distance, after the end of supplementation, both in the supplemented group as well as in the total group of players. 

Significant changes in VO2max have been noted in soccer players after 8 weeks of supplementation, although the dose of vitamin D was lower than in the present study [69]. Similarly, vitamin D supplementation demonstrated a positive effect on aerobic capacity among rowers undergoing high-intensity training [68]. Several studies have found a positive significant correlation between VO2max and vitamin D concentration in professional soccer players [81]. In addition, a similar correlation has been documented in an untrained person [84]. The mechanism regulating the relation between the 25(OH)D concentration and the level of aerobic capacity is still not fully recognized. Previous studies demonstrated that vitamin D may support the erythropoiesis process [85], which results in larger number of red blood cells. The increased number of these oxygen transporters could positively affect the level of VO2max. This issue should be further explored in future research concerning the association between vitamin D and aerobic metabolism during exercise.

In our study, a significant, fair, positive correlation was found between VO2max and 25(OH)D concentration for the TG. This dependence may suggest that players with a higher level of 25(OH)D concentration may have better aerobic fitness. However, some previous studies indicate no potential relationship between 25(OH)D level and VO2max [67,86], nor any significant effect of vitamin D supplementation on athletic aerobic performance [39,48]; in addition, an opposite correlation to our present findings regarding VO2max and vitamin D concentrations have been noted in a group of footballers [67]. This variation is not surprising considering that cardiovascular fitness and the VO2max parameter (the gold standard for evaluation) depend on a range of factors, including heredity, age, gender, and body composition, as well as various physiological aspects, such as blood oxygen capacity, circulating blood volume, cardiac output, pulmonary blood flow, lung diffusion capacity, and lung ventilation [83].

Therefore, the effect of vitamin D supplementation on performance depends on a multitude of factors, including training components. A recently published meta-analysis on vitamin supplementation (2850–5000 IU vitamin D for over 4 weeks) found it to have no significant overall effect on global muscle strength, mass, and muscle power, or at least a small effect size [32]. Our present findings are in line with this meta-analysis data. The results of published studies on the effect of vitamin D supplementation on skeletal muscle function and physical performance are difficult to compare due to the small group of athletes and various sports disciplines, as well as the different geographic conditions of the athletes’ residence. The differences in the applied training loads should also be pointed out. 

The experimental study design is undoubtedly a strength of this research. Another is the possibility to apply the treatment during the winter months, a period in which the limited exposure to sunlight allows a clearer picture of the effect of dietary vitamin D, and the fact that this research was performed in training athletes. 

We are aware that our study contains some limitations, mainly due to the small group of examined participants; this was exacerbated by the fact that only 63% of players were included in the analysis due to injuries or transfers. Additionally, despite the fact that our population included young and healthy athletes, there is a possibility that individuals may have suffered from asymptomatic diseases that may significantly affect vitamin D absorption. Finally, due to the fact that body fat content is linked with vitamin D redistribution, the data of players’ body composition (especially fat percentage) would undoubtedly enrich the data interpretation.

## 5. Conclusions

The present study is one of the few performed on young athletes. Previous research did not establish a clear consensus regarding the effect of serum vitamin D concentration (and supplementation) on speed and strength/power performance in athletes. Our results indicate that the applied vitamin D supplementation, in a dosage of 20,000 IU of cholecalciferol twice a week for 8 weeks, is related primarily to its ergonomic function. It was found that vitamin D supplementation increased the 25(OH)D serum level in the young soccer players. However, according to previous studies [87], such doses should be used in other age groups with caution. Moreover, in the TG, the 25(OH)D concentration was significantly and positively correlated to the level of aerobic power; however, it was not associated with a clear improvement in muscle strength/power. It is very likely that a lower dose (similar to that used in untrained people) may improve players’ physical performance when a deficiency or suboptimal concentration of the above vitamin is restored. Nevertheless, vitamin D supplementation should be recommended to athletes, especially during periods of intense training and during winter–spring, periods associated with vitamin D deficiency/insufficiency.

## Figures and Tables

**Figure 1 ijerph-19-05138-f001:**
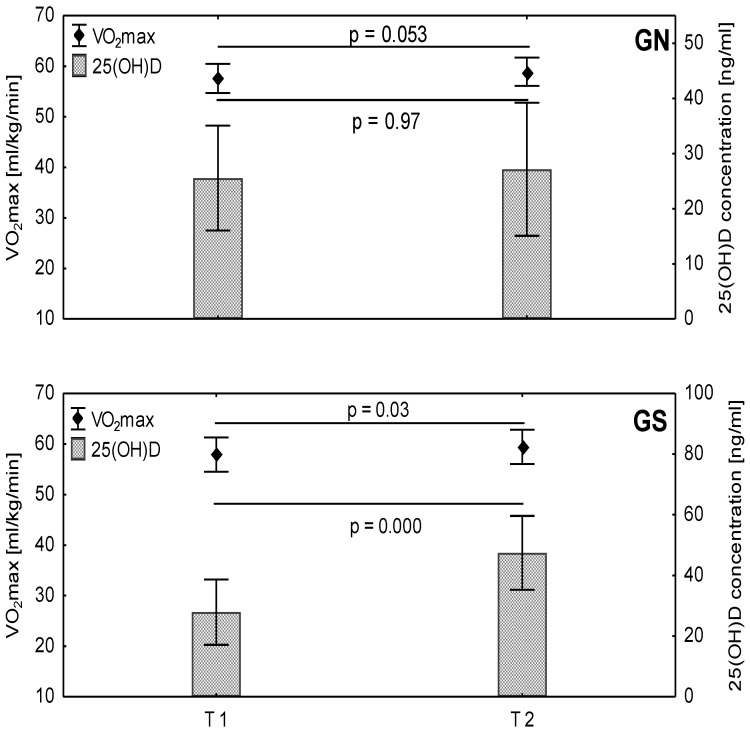
Changes of maximal oxygen uptake (VO2max) and 25(OH)D concentration during the experimental period in the non-supplemented group (GN) and the supplemented group (GS).

**Figure 2 ijerph-19-05138-f002:**
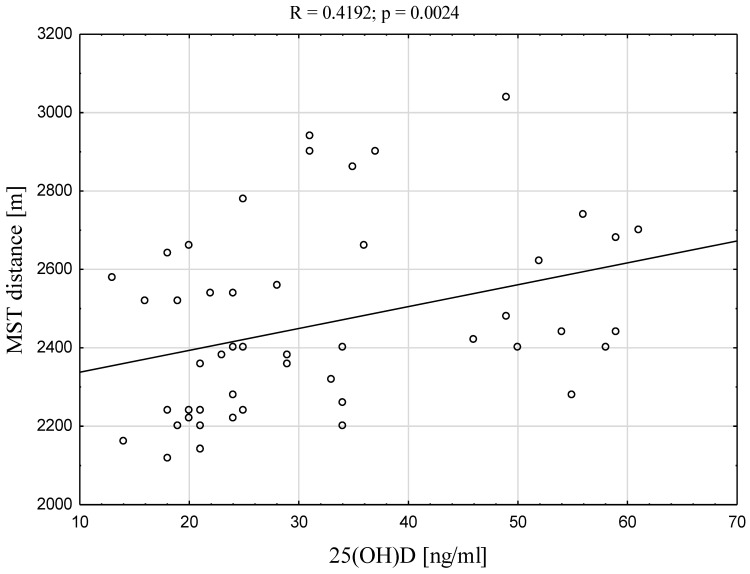
Correlation between of 25(OH)D concentration and MST distance and 25(OH)D and VO2max in young soccer players.

**Table 1 ijerph-19-05138-t001:** Selected biological features of soccer players included in the study and selected training characteristics.

Parameter	Total Group (TG)
Age [years]	17.5 ± 0.70
Body height [cm]	Height 178 ± 0.70
Body weight [kg]	Weight 68.05 ± 9.18
Training period [weeks]	8 weeks in the preparation period (winter, from mid-January to mid-March)
Training unit [min]	Each training session was a continuous 90 min, 5 training units, and a control (friendly) match each week. In addition, the players participated in 2 units of physical education lessons at school (90 min) with a focus on soccer practice.

**Table 2 ijerph-19-05138-t002:** The 25(OH)D concentrations and the physical fitness values in the tested athletes: total group (TG), the group supplemented with vitamin D (GS), and the non-supplemented group (GN) at baseline (T1) and after 8 weeks (T2).

Group	TG		GN		GS		Interactions	*p*	ES
Time-Point	T1	T2	T1	T2	T1	T2			
25(OH)D [ng/mL]	26.7 ± 10.01	36.9 ± 15.74 †	25.5 ± 9.52	27.2 ± 12.06	27.9 ± 10.79	47.4 ± 12.21 *†	group time group × time	0.006 0.0003 0.002	0.29 0.44 0.36
Sprint 5 m [s]	1.01 ± 0.04	1.02 ± 0.05	1.01 ± 0.06	1.01 ± 0.05	1.01 ± 0.03	1.02 ± 0.05	group time group × time	0.80 0.14 0.36	0.00 0.09 0.04
Sprint 10 m [s]	1.77 ± 0.06	1.77 ± 0.06	1.77 ± 0.07	1.75 ± 0.06	1.77 ± 0.06	1.78 ± 0.06	group time group × time	0.45 0.50 0.05	0.02 0.02 0.15
Sprint 30 m [s]	4.24 ± 0.13	4.23 ± 0.13	4.23 ± 0.18	4.21 ± 0.15	4.25 ± 0.12	4.25 ± 0.10	group time group × time	0.54 0.48 0.57	0.02 0.02 0.01
MST dist. [m]	2408 ± 226.0	2510 ± 227.1 †	2395 ± 210.9	2492 ± 208.7	2422 ± 250.0	2530 ± 253.3 †	group time group × time	0.72 0.000 0.83	0.01 0.42 0.00
VO2max [ml/kg/min]	57.7 ± 3.08	59.2 ± 3.05 †	57.6 ± 2.88	58.9 ± 2.82	57.9 ± 3.40	59.4 ± 3.39 †	group time group × time	0.73 0.001 0.80	0.01 0.41 0.00
10 jumps [cm]	37.4 ± 3.55	38.5 ± 3.45	38.5 ± 3.33	39.4 ± 2.66	36.1 ± 3.49	37.6 ± 4.05	group time group × time	0.09 0.07 0.62	0.12 0.13 0.01
10 jumps [W/kg]	48.3 ± 3.07	49.3 ± 2.92	49.4 ± 2.97	50.0 ± 2.33	47.2 ± 2.86	48.4 ± 3.36	group time group × time	0.07 0.10 0.58	0.13 0.11 0.01
SJ [cm]	36.0 ± 4.09	36.6 ± 4.41	36.6 ± 3.99	37.5 ± 4.46	35.3 ± 4.27	35.6 ± 4.33	group time group × time	0.33 0.28 0.61	0.04 0.05 0.01
SJ [W/kg]	47.1 ± 3.55	47.6 ± 3.73	47.8 ± 3.50	48.4 ± 3.76	46.4 ± 3.62	46.8 ± 3.67	group time group × time	0.30 0.32 0.73	0.05 0.04 0.01
CMJ [cm]	43.9 ± 4.57	44.0 ± 3.94	45.4 ± 4.8	44.7 ± 4.17	42.3 ± 3.88	43.2 ± 3.67	group time group × time	0.14 0.86 0.26	0.09 0.00 0.05
CMJ [W/kg]	53.9 ± 4.26	53.9 ± 3.45	55.3 ± 4.80	54.6 ± 3.75	52.4 ± 3.14	53.1 ± 3.07	group time group × time	0.13 0.96 0.25	0.09 0.00 0.06

* significantly different from GN; ^†^ significantly different from T1. TG—total group, GN—non-supplemented group, GS—supplemented group, 25(OH)D—serum concentrations of 25(OH)D, SJ—squat jump, CMJ—countermovement jump, MST dist.—multistage shuttle run test.

**Table 3 ijerph-19-05138-t003:** Correlation between 25(OH)D concentration and physical fitness parameters in young soccer players.

Variables	25(OH)D [ng/mL]	
	R	*p*-Value
Sprint 5 m [s]	0.0562	0.6981
Sprint 10 m [s]	0.0138	0.9242
Sprint 30 m [s]	0.0920	0.5251
MST dist. [m]	0.4192	0.0024 *
VO2max [ml/kg/min]	0.4192	0.0024 *
10 jumps [cm]	−0.1878	0.1916
10 jumps [W/kg]	−0.1937	0.1777
SJ [cm]	−0.0680	0.6390
SJ [W/kg]	−0.0729	0.6149
CMJ [cm]	−0.0960	0.5074
CMJ [W/kg]	−0.0997	0.4910

* statistically significant.

## Data Availability

The data presented in this study are available on request from the corresponding author upon reasonable request.

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
