# Peer review of "Correlation between the Positive Effect of Vitamin D Supplementation and Physical Performance in Young Male Soccer Players"

_ijerph, 2022, doi:10.3390/ijerph19095138_

Round 1

Reviewer 1 Report

The work presented in this manuscript compared the performance in a number of selected tests before and after a four-week vitamin D supplementation regimen, in a group of young soccer athletes (n=25). The athletes were randomly assigned to a supplementation group (20000 IU vitamin D twice a week) and a control group (no supplementation) in a double-blind manner (as stated by the authors). The authors reported that the athletes demonstrated a deficiency of vitamin D and the supplementation improved it. The supplementation showed an ergonomic effect on VO2max but not on jump and sprint performance.

The manuscript is well presented in general, and the findings could make valuable contributions to the field if some methodological issues could be clarified and explained further. I would provide the following critiques for authors’ consideration, aiming to improve the quality of the manuscript.

One major concern I have is the so claimed double-blind design. It is not clear (not explained) whether the GN group was given any placebo at the same time. If not, the participants would have known who was given the supplement or not, that would breach the blind design, and might have an effect on the outcomes of the study. Please clarify this issue.

Line 1-3: The title should reflect that the participants were young (under 18 on average) male athletes.

Line 92-99: The rationale for this study was an assumed global vitamin D deficiency in athletes population, especially those who participate in indoor training and/or with sun deficiency. However, the authors did not explain whether this group of young soccer players had these conditions. (It was addressed later in the Discussion, but should be addressed here to support the rationale).

The Introduction lacks of background information on the roles of Vit D in the body, particularly the possible mechanism/s for the claimed beneficial effect of Vit D supplementation. The background information should focus more on athletes, and not be a mixture of reports on patients, healthy individuals and athletes (with and without Vit D deficiency).

Line 141-148: How was the blindness ensured to athletes? For example, was the GN given a placebo while GS was given the Vit D supplement, that prevented them from knowing which group they were in?

Line 144: How was the dosage determined (with reference)?

Line 183-186: need to cite reference/s for the reference values. Were the values for healthy male adults or more specifically for children/adolescents or athletes?

Line 223: Did the authors use a mixed ANOVA, as the between-group comparison was not a repeated measure?

Line 289-292: These should be presented in Introduction to strengthen the rationale for this study.

Line 334-335: I am not sure if the doses given twice a week could be presented as daily dose. It would be appreciated by the readers if the dynamics of Vit D supplement were explained (e.g. the rate of absorption and metabolism, half-life in blood and tissues, etc.).

Also, please explain the reason for the regimen of twice a week but not daily supplementation.

Line 399-405: Readers would appreciate some explanation or speculation on the possible mechanisms for improved VO2max after Vit D supplementation. How would the exogenous Vit D affect the factors listed here?

Line 425: I would not agree with this statement if no placebo was given to the GN.

Line 439-440: as questioned above, it might not be appropriate to present the dosage as a daily dose.

Line 441-445: the statement is confusing. It reads like both VO2max and muscle strength/power were not correlated with the Vit D level or not improved with the supplementation. Please revise.

Reviewer 2 Report

The aim of this study is to determine whether supplementation of vitamin 97 D (20 000 IU of cholecalciferol twice a week for 8 weeks) influence muscle strength/power and aerobic capacity in young soccer players.

As shown in the introduction, vitamin D supplementation and its influence on sports performance has been extensively studied in the scientific literature and it is difficult to add to the studies already carried out to date.

On the other hand, the way in which the introduction has been written, focusing initially on vitamin D rather than on aspects or variables related to sports performance, does not seem to be the most appropriate way to introduce the paper for a journal on sport and health.

Here are my contributions:

- In table 1 it appears that the study was carried out during the preparatory period, but in the same table and in line 118 it is stated that during this period they are playing a match on weekends. In which period was the study carried out? if they played a match at the weekend, did all players play the match? did they all bear the same training load? of each group carried out for the study, how many players of each group played the full matches?

  • Related to the above, line 126 comments that only subjects who performed 85% of the training and matches were considered, but 25 subjects are included in the study. How can it be that by playing 11 players on the field, with some change available, the 25 players can fulfil the total of 85% of the training sessions AND MATCHES? furthermore, there were no injuries in that period?
  • Line 132, how did participants demonstrate adherence to the diet? is a variable that can greatly influence the results obtained.
  • Line 148, was any placebo provided to the group that did not take vitamin D?
  • Line 209, considering the number of subjects included in the study, the estimation of VO2max by the test used is hardly justifiable.
  • Line 306, add some reference.
  • The first 5 paragraphs of the discussion only refer to the reasons for the study and the improvement of 25(OH)D levels in the supplemented subjects. This result is not the most relevant of the study and is not very difficult to predict, so there is no point in such a length of text.
  • Line 390, in addition to comparing one study with another, what do you think are the reasons why there are so many different results in the studies that have been carried out, and to delve more deeply into the reasons why some studies contradict others and why your results may or may not coincide with the other research.

Reviewer 3 Report

The authors present an interesting aspect of supplementation applied to sport, in particular, related to vitamin D

However, there are some points to be clarified/improved:

- do you have data on body composition available? Plicometry, bioimpedance? I guess an iDXA is unfortunate given the level; it would be important since both the general state and the percentage of fat are linked to the redistribution of vitamin D

- Given the high variance it would be better to divide the athletes into two groups (even if the sample is limited), as the value of 26 is to be considered insufficient, while 16 would be an important deficiency

- The proposed dosages are high, in fact, the average reached beyond what is often considered a good value, that is between 30 and 40, especially considering the variance that would bring some values almost to 60

On the other hand, I fully agree with the advisability of supplementation, particularly in winter.

Round 2

Reviewer 1 Report

Authors have adequately addressed most of my previous comments and revised the manuscript accordingly. I appreciate their efforts in revising the manuscript with detailed responses/explanation.

However, there are still some areas that needs clarification in the revised manuscript. A thorough proof reading is also needed.  

Line 19: “eight weeks of”

Line 21: “A set of measurements…… was…..”

Line 25-26: “A significant improvement in VO2max was found in …”.

Line 71: “…including those in bones and muscles”….”It has been recognized ……acts…”

Line 78: should indicate that the claimed effect on muscle cell proliferation and differentiation was found in animal studies.

Line 81: “one animal…”

Line 84: use “key factors” or “a key factor”

Line 85: “may improve aerobic performance”

Line 86: please spell out CYP

Line 94: “is ubiquitous”

Line 287-294: Good to see the explanation on how the double-blind design was carried out. However, how about the researchers who analysed the data? Did they know who were in which group before the data analysis?

Line 506: “… in some studies…”

Line 606-629: if this is a speculation, authors may indicate further research is test this speculation or hypothesis.

Line 667: should use “correlated” rather than “related”

Conclusions: The issue about the VO2max is still confusing. It is not clear whether the significant correlation between Vit D and VO2max was found in the TG, GS or GN. If it was for the TG, then the effect of the intervention could not be properly identified (the difference between the GS and GN). Please make the point clearer.

Further, VO2max represents aerobic power, rather than capacity.

Line 667: “correlated”

Reviewer 2 Report

Despite considerable improvements in the manuscript, the methodology (measurement of VO2max, quantification of the participants' load...) still has deficiencies.

Reviewer 3 Report

Even though I agree with the supplementation of vitamin D, particularly in Northern Europe, the dosage is unjustifiably higher, the literature cited is not enough nor conclusive to assess that is the best way and the value is quite high.

So it is a dangerous message that it could be passed the more is the best; our effort should be to communicate the right dosage following the concept of hormesis, I think that is mandatory to include a statement that even you choose a dosage of 5000IU it is very likely that a lower dosage (close to that used for normal population) should be used because vitamin D doesn't show any ergogenic power per se; it can improve performance when a deficiency or a non-optimum value is restored.

The author doesn't make any change following my suggestions so my opinion is the same....
